# Provably Secure Three-Factor-Based Mutual Authentication Scheme with PUF for Wireless Medical Sensor Networks

**DOI:** 10.3390/s21186039

**Published:** 2021-09-09

**Authors:** DeokKyu Kwon, YoHan Park, YoungHo Park

**Affiliations:** 1School of Electronic and Electrical Engineering, Kyungpook National University, Daegu 41566, Korea; kdk145@knu.ac.kr; 2School of Computer Engineering, Keimyung University, Daegu 42601, Korea; yhpark@kmu.ac.kr; 3School of Electronics Engineering, Kyungpook National University, Daegu 41566, Korea

**Keywords:** wireless medical sensor networks, PUF, biometrics, BAN logic, RoR model, AVISPA

## Abstract

Wireless medical sensor networks (WMSNs) are used in remote medical service environments to provide patients with convenient healthcare services. In a WMSN environment, patients wear a device that collects their health information and transmits the information via a gateway. Then, doctors make a diagnosis regarding the patient, utilizing the health information. However, this information can be vulnerable to various security attacks because the information is exchanged via an insecure channel. Therefore, a secure authentication scheme is necessary for WMSNs. In 2021, Masud et al. proposed a lightweight and anonymity-preserving user authentication scheme for healthcare environments. We discover that Masud et al.’s scheme is insecure against offline password guessing, user impersonation, and privileged insider attacks. Furthermore, we find that Masud et al.’s scheme cannot ensure user anonymity. To address the security vulnerabilities of Masud et al.’s scheme, we propose a three-factor-based mutual authentication scheme with a physical unclonable function (PUF). The proposed scheme is secure against various security attacks and provides anonymity, perfect forward secrecy, and mutual authentication utilizing biometrics and PUF. To prove the security features of our scheme, we analyze the scheme using informal analysis, Burrows–Abadi–Needham (BAN) logic, the Real-or-Random (RoR) model, and Automated Verification of Internet Security Protocols and Applications (AVISPA) simulation. Furthermore, we estimate our scheme’s security features, computation costs, communication costs, and energy consumption compared with the other related schemes. Consequently, we demonstrate that our scheme is suitable for WMSNs.

## 1. Introduction

With the development of wireless communication and sensor minimization technology, wireless sensor networks (WSNs) have been widely used in various environments, such as industrial Internet of Things [1], healthcare [2], and smart homes [3]. In particular, the demand for remote healthcare services has been increased due to the COVID-19 pandemic [4]. Remote healthcare services can be realized through wireless medical sensor networks (WMSNs). Generally, WMSNs consist of doctors (users), a gateway, and sensor nodes. Doctors communicate with the gateway to access a patient’s health data through their smart device. The gateway, such as a smart hospital, stores sensitive data and supports smooth wireless communication between doctors and sensor nodes. Sensor nodes are attached to patients and transmit patients’ sensitive health data to doctors through the gateway [5]. Therefore, doctors can perform the diagnosis of patients remotely and patients can receive convenient remote medical services wherever they are.

Although WMSNs can provide convenient medical services to patients, there are several security risks. First of all, each message is exchanged through a public channel; therefore, malicious adversaries can perform security attacks such as replay and man-in-the-middle attacks [6]. In addition, the smart device of a doctor can be stolen and an adversary can attempt to impersonate the doctor using parameters extracted from the device. In addition, the sensor node can be physically captured by an adversary and the adversary can attempt to impersonate the patient using the secret parameter, extracted from the sensor node. If an adversary obtains and modifies the information of patients using the above security attacks, this can have a serious effect on the patient’s health, such as inducing a misdiagnosis by the doctor. Accordingly, secure authentication schemes are necessary to overcome these security vulnerabilities for WMSNs.

In 2021, Masud et al. [7] proposed a lightweight and anonymity-preserving user authentication scheme for IoT-based healthcare environments. They claimed that their scheme is lightweight and prevents various security attacks (e.g., replay, privileged insider, and impersonation attacks). Moreover, they asserted that their scheme can ensure user anonymity and session key agreement. However, we find that Masud et al.’s scheme cannot prevent offline password guessing, user impersonation, and privileged insider attacks. Moreover, we prove that their scheme cannot ensure user anonymity. Their scheme also has a device update problem, where the doctor cannot perform a login process on his own smart device. To overcome these security vulnerabilities of Masud et al.’s scheme, we propose a secure three-factor-based mutual authentication scheme with physical unclonable function (PUF) for WMSNs. In our scheme, we use PUF and fuzzy extractor [8] to enhance the security level. The PUF is a physical circuit that outputs unpredictable random strings, and the fuzzy extractor is a cryptographic algorithm that utilizes the biometrics of users. Therefore, we install the PUF in the sensor node to prevent physical and cloning attacks, and we utilize the fuzzy extractor to overcome offline password guessing attacks. Our scheme also uses hash functions and exclusive-OR operations to ensure real-time communication.

### 1.1. Research Contributions

The contributions of our paper are as follows.

We review Masud et al.’s scheme and prove that their scheme cannot ensure user anonymity. Moreover, we show that their scheme is vulnerable to offline password, impersonation, and privileged insider attacks and has a device update problem.We propose a secure three-factor-based mutual authentication scheme to overcome the security vulnerabilities of Masud et al.’s scheme. We use hash functions and exclusive-OR operations to provide real-time communication for WMSNs. We also utilize PUF and fuzzy extractor [8] to prevent physical and offline password guessing attacks, respectively.We analyze the security features of the proposed scheme using well-known Burrows–Abadi–Needham (BAN) logic [9] and the Real-or-Random (RoR) model [10], which can prove mutual authentication and session key security, respectively. Furthermore, we utilize the Automated Verification of Internet Security Protocols and Applications (AVISPA) simulation tool [11,12] to prove that the proposed scheme has resistance against replay and man-in-the-middle attacks.We show that our scheme has resistance against various security attacks, such as offline password, impersonation, privileged insider, replay, and man-in-the-middle attacks, using informal analysis. Moreover, the proposed scheme ensures user anonymity, perfect forward secrecy, and mutual authentication.We estimate the security properties and functionalities, communication costs, computation costs, and energy consumption of our scheme in comparison with existing authentication schemes.

### 1.2. Organization

In Section 2, we introduce related works for WMSNs. We describe the PUF, fuzzy extractor, adversary model, and system model in Section 3. In Section 4, we describe the detailed procedures of Masud et al.’s scheme. In Section 5, we prove the security vulnerabilities of Masud et al.’s scheme. To overcome these security vulnerabilities, we propose a secure three-factor-based mutual authentication scheme with PUF for WMSNs in Section 6. In Section 7 and Section 8, we analyze the security features of our scheme using formal and informal analyses and estimate the performance of our scheme, respectively. Finally, we conclude and summarize our paper in Section 9.

## 2. Related Works

In the past several decades, researchers have proposed numerous two-factor-based authentication schemes for WMSNs. In 2012, Kumar et al. [13] proposed an authentication scheme for healthcare applications using a smart card. Their scheme used a symmetric encryption method to establish the session key between the user and the medical sensor node. However, He et al. [14] claimed that Kumar et al.’s scheme is vulnerable to password guessing and privileged insider attacks. As a result, He et al. proposed a robust authentication scheme to overcome these security weaknesses. Unfortunately, Mir et al. [15] demonstrated that [14] cannot prevent offline password guessing and masquerading user attacks. To address the security vulnerabilities of He et al’s scheme [15], they proposed an authentication and key agreement scheme using hash functions and exclusive-OR operations. In 2018, Wu et al. [16] proposed an authentication scheme for personalized healthcare systems. They used a smart device as a factor to protect the privacy of the doctor. However, the above schemes [13,14,15,16] can be vulnerable to smart device theft and offline password guessing attacks because they adopt two-factor-based authentication schemes.

Three-factor-based authentication schemes have been proposed to improve the security level for WMSNs. In 2018, Challa et al. [17] proposed a three-factor-based user authentication and key agreement protocol using bilinear pairings for wireless healthcare sensor networks. Challa et al. employed bilinear pairing and the fuzzy extractor to overcome security vulnerabilities such as smart card theft, offline password guessing, and privileged insider attacks. In 2019, Li et al. [18] proposed a three-factor user authentication protocol based on elliptic curve cryptography (ECC). They claimed that their scheme can resist various security attacks utilizing biometrics verification with error-correcting code and a fuzzy commitment scheme. Shin et al. [19] suggested an authentication and key agreement scheme that can preserve users’ privacy in 5G-integrated IoT environments. In [19], each entity establishes the session key using elliptic curve Diffie–Hellman (ECDH). Furthermore, Ali et al. [20] proposed a biometric-based authentication and access control protocol for WMSNs using ECC. They claimed that their scheme is secure against privileged insider, stolen smart card, and offline password guessing attacks. In 2020, Hsu et al. [21] proposed a three-factor user-controlled single sign-on (UCSSO) scheme for telecare medicine information systems. Their scheme can provide fast authentication and privacy protection using only hash functions and exclusive-OR operations. Although the above schemes [18,19,20,21] can provide lightweight communications to doctors and patients, they cannot prevent sensor node physical and cloning attacks.

Recently, PUF-based authentication schemes have been proposed to prevent physical attacks. In 2017, Aman et al. [22] suggested a mutual authentication scheme using PUF in IoT systems. They claimed that their scheme is secure against IoT device cloning attacks because PUF is employed on each IoT device. Byun [23] proposed an end-to-end key exchange scheme using PUF. This scheme utilized PUF-embedded devices and the fuzzy extractor to ensure mutual authentication between two devices. In 2020, Fang et al. [24] proposed a PUF-based data transmission scheme for IoT environments. They proved that their scheme can prevent various attacks, such as DoS, eavesdropping, impersonation, and cloning attacks, using PUF. In 2021, Chen et al. [25] suggested an efficient mutual authentication and key agreement scheme using PUF and biometrics for wireless sensor network environments. To reduce the storage overhead of the user, Chen et al. [25] eliminated the password during the login phase.

In 2021, Masud et al. [7] proposed a lightweight user authentication scheme for IoT-based healthcare. They asserted that their scheme can protect against impersonation attacks and replay attacks and provide data privacy and anonymity. However, we discover that their scheme is vulnerable to several security issues, such as offline password guessing, user impersonation, and privileged insider attacks. We also find that their scheme cannot ensure user anonymity. Therefore, we propose a three-factor-based mutual authentication scheme using PUF to prevent various security weaknesses such as user anonymity, smart device theft, offline password, privileged insider, and cloning attacks, which are critical for WMSNs.

## 3. Preliminaries

In this section, we introduce the general system model and the adversary model for WMSNs. Then, we describe PUF and the fuzzy extractor, which can improve the security level of our scheme.

### 3.1. System Model

Figure 1 shows the general system model of a WMSN, which consists of doctors, a gateway, and sensor nodes. Details are as follows.

Doctor (user): The doctor, who has a resource-constrained smart device, authenticates with the gateway to access patients’ health reports. To communicate with sensor nodes, the doctor must register with the gateway.Gateway: The gateway, which is the smart hospital, communicates with doctors and sensor nodes to provide efficient and convenient remote services to patients. We assume that the gateway is a trusted party and has enough storage and computing power.Sensor node: The sensor node is a resource-constrained device attached to the patient in the form of a wearable device. The sensor node collects the patient’s health information and sends it to the doctor through the gateway.

### 3.2. Adversary Model

In our paper, we assume that an adversary can eavesdrop, insert, remove, and modify messages transmitted through a public channel according to a well-known adversary model, the Dolev–Yao (DY) model [26]. Moreover, we use the Canetti–Krawczyk (CK) adversary model [27]. In this model, an adversary can access ephemeral parameters or the master key of the gateway. With the CK and DY adversary models, we assume that an adversary can perform various attacks. Details are as below.

An adversary can steal a doctor’s smart device and obtain the secret parameter, extracted from the smart device using a power analysis attack [28].An adversary can be a privileged insider who can obtain the user’s registration message.An adversary can obtain the patient’s sensor node and perform a cloning attack.An adversary can perform various attacks, such as man-in-the-middle, password guessing, and stolen verifier attacks [29].

### 3.3. Physical Unclonable Function

Physical unclonable functions (PUFs) are physical circuits that operate as a one-way function. In the PUF circuit, there is an input–output bit string pair called the “challenge–response pair”. If a random bit string challenge is entered into the PUF circuit, a unique output response is printed out. In this paper, we express this process as R=PUF(C), where *C* and *R* are a challenge and a response, respectively. Ideal PUF properties are as below.

The PUF is an unclonable circuit.The PUF is a unique physical microstructure. The output of the PUF depends on the physical circuit.The output of the PUF has to be unpredictable.The circuit of the PUF is easy to estimate and implement.

Since a PUF has the properties of a one-way function, the PUF returns the same response when the same challenge is input into a PUF-installed device. Moreover, the PUF gives different responses when the same challenge is input into different devices. Therefore, the PUF can provide a unique one-way function that cannot be duplicated. This uniqueness enables the PUF to prevent various attacks, such as physical and cloning attacks.

### 3.4. Fuzzy Extractor

In this section, we explain the basic concept and direction of the fuzzy extractor [8]. When a user utilizes his biometrics or the PUF response string, we cannot ensure the accuracy due to the noise of external environmental factors. The fuzzy extractor can control the noise using the helper string. Therefore, we can use the biometric information and the PUF response string as a secret parameter using the fuzzy extractor. The fuzzy extractor consists of “generate (Gen(.))” and “reproduce (Rep(.))” algorithms. Details are as follows.

$Gen\left(B_{i}\right)=\left(R_{i},P_{i}\right)$: This is a probability algorithm to generate a secret string $R_{i}$. If a user inputs a random string $B_{i}$, the fuzzy extractor generates the secret parameter $R_{i}$ and a helper string $P_{i}$.$Rep\left(B_{i}^{*},P_{i}\right)=\left(R_{i}\right)$: This is a deterministic algorithm to reproduce the secret string $R_{i}$. If a user enters the random string $B_{i}^{*}$, the fuzzy extractor controls the noise of $B_{i}^{*}$ using the helper string $P_{i}$ and reproduces the secret string $R_{i}$.

## 4. Review of Masud et al.’s Scheme

In 2021, Masud et al. [7] proposed a lightweight and anonymity-preserving user authentication scheme for IoT-based healthcare environments. Their scheme consists of user registration, sensor node registration, and mutual authentication and key agreement phases. Notations and descriptions are explained in Table 1.

### 4.1. User Registration Phase

A doctor must register in the gateway to use this network system. We show the user registration phase of Masud et al.’s scheme as follows.

**Step 1:** The doctor inputs an identity $D_{ID}$ and password $PW_{D}$, and generates a registration request message $R_{req}$. Then, the doctor sends $M_{RD}^{1}=\left\{D_{ID},PW_{D},R_{req}\right\}$ to the gateway through a secure channel.

**Step 2:** The gateway stores $D_{ID}$ and $PW_{D}$, and then generates $R_{SG}^{1}$ to compute $\alpha =\left(D_{ID}⊕R_{SG}^{1}\right)⊕PW_{D}$ and $D_{TID}=R_{SG}^{1}⊕D_{ID}$. The gateway stores $\left\{R_{SG}^{1},D_{TID}\right\}$ in its secure database and sends α to the doctor via a secure channel.

**Step 3:** The doctor computes $R_{SG}^{1*}=\left(\alpha ⊕PW_{D}\right)⊕D_{ID}$ and $D_{TID}=R_{SG}^{1*}⊕D_{ID}$, and stores $\left\{R_{SG}^{1*},D_{TID}\right\}$ in his device. Then, the doctor computes $\beta =h(PW_{D}\left|\right|R_{SG}^{1*})⊕D_{TID}$ and stores {β}.

### 4.2. Sensor Node Registration Phase

To transmit the health information of a patient, the sensor node must register with the gateway. We describe the sensor node registration phase as below.

**Step 1:** The sensor node generates $R_{SN}^{1}$, and sends $\left\{S_{ID},R_{SN}^{1}\right\}$ to the gateway via a secure channel, where $S_{ID}$ is the real identity of the sensor node.

**Step 2:** The gateway generates $R_{SG}^{2}$ and computes $\delta =\left(S_{ID}⊕R_{SG}^{2}\right)⊕R_{SN}^{1}$ and $S_{TID}=R_{SG}^{2}⊕S_{ID}$. Then, the gateway stores $\left\{S_{ID},R_{SN}^{1},R_{SG}^{2},S_{TID}\right\}$ in its secure database and transmits {δ} to the sensor node through a secure channel.

**Step 3:** When the sensor node receives {δ}, it computes $R_{SG}^{2*}=\left(\delta ⊕R_{SN}^{1}\right)⊕S_{ID}$ and $S_{TID}=R_{SG}^{2*}⊕S_{ID}$. Finally, the sensor node stores $\left\{R_{SN}^{1},R_{SG}^{2*},S_{TID}\right\}$ in its memory.

### 4.3. Mutual Authentication and Key Agreement Phase

In this phase, the doctor and the sensor node conduct a mutual authentication and key agreement phase to authenticate each other and establish a session key. Figure 2 shows the mutual authentication and key agreement phase of Masud et al.’s scheme and details are as follows.

**Step 1:** When the doctor inputs his own password $PW_{D}$, the smart device of the doctor computes $Q=h(PW_{D}||R_{SG}^{1*})$ and verifies $Q\overset{?}{=}\;\beta$. If it is correct, the smart device generates a random nonce $N_{D}^{1}$ and computes $N_{D}^{1*}=N_{D}^{1}⊕PW_{D}$ and $\lambda =h(R_{SG}^{1*}||PW_{D})$. Then, the doctor sends $\left\{N_{D}^{1*},D_{TID},\lambda ,S_{TID}\right\}$ to the gateway via a public channel.

**Step 2:** The gateway receives $\left\{N_{D}^{1*},D_{TID},\lambda ,S_{TID}\right\}$ and computes $N_{D}^{1}=N_{D}^{1*}⊕PW_{D}$. If $N_{D}^{1}$ is a fresh random nonce, the gateway checks the validity of $S_{TID}$ and $D_{TID}$, and computes $\lambda^{*}=h(R_{SG}^{1}||PW_{D})$. After verifying the equation $\lambda^{*}\overset{?}{=}\lambda$, the gateway generates $N_{G}^{1}$ and computes $G_{W}^{1}=N_{G}^{1}⊕S_{TID}$, $G_{W}^{2}=h(R_{SN}^{1}\left|\right|R_{SG}^{2})$, $SK_{S}=\left(SK⊕R_{SN}^{1}\right)⊕N_{G}^{1}$, and $G_{W}^{3}=R_{SG}^{3}⊕R_{SN}^{1}$, where SK is a session key. Then, the gateway sends $\left\{G_{W}^{1},G_{W}^{2},D_{TID},SK_{S},G_{W}^{3}\right\}$ to the sensor node through a public channel.

**Step 3:** The sensor node computes $N_{G}^{1}=G_{W}^{1}⊕S_{TID}$ and checks the freshness of $N_{G}^{1}$. After this, the sensor node computes $S_{N}^{1}=h(R_{SN}^{1}\left|\right|R_{SG}^{2})$ and checks the equality of $S_{N}^{1}$ and $G_{W}^{2}$. If it is equal, the sensor node generates $N_{S}^{1}$ and computes $SK=\left(SK_{S}⊕N_{G}^{1}\right)⊕R_{SG}^{1}$, $S_{N}^{2}=N_{S}^{1}⊕S_{TID}$, $S_{N}^{3}=h(R_{SG}^{2*}\left|\right|R_{SN}^{1}\left||SK\right)$, $S_{N}^{4}=R_{SG}^{2}⊕R_{SN}^{2}$, $R_{SG}^{3}=G_{W}^{2}⊕R_{SN}^{1}$, and $S_{TID}^{new}=R_{SG}^{3}⊕S_{ID}$. Finally, the sensor node stores $\left\{R_{SN}^{2},R_{SG}^{3},S_{TID}^{new}\right\}$ and transmits $\left\{S_{N}^{2},S_{N}^{3},S_{N}^{4}\right\}$ to the gateway.

**Step 4:** When the gateway receives $\left\{S_{N}^{2},S_{N}^{3},S_{N}^{4}\right\}$ from the sensor node, the gateway computes $N_{S}^{1}=S_{N}^{2}⊕S_{TID}$ and verifies the freshness of $N_{S}^{1}$. Then, the gateway computes $G_{W}^{4}=h(R_{SG}^{2}\left|\right|R_{SN}^{1}\left||SK\right)$ and checks $G_{W}^{4}\overset{?}{=}S_{N}^{3}$. If it is equal, the gateway computes $R_{SN}^{2}=S_{N}^{4}⊕R_{SG}^{2}$ and $S_{TID}^{new}=R_{SG}^{3}⊕S_{ID}$ and stores $\left\{R_{SN}^{2},R_{SG}^{3},S_{TID}^{new}\right\}$ in its database. The gateway generates a random nonce $N_{G}^{2}$ and computes $\mu =D_{ID}⊕N_{G}^{2}$, $SK_{U}=\left(SK⊕PW_{D}\right)⊕N_{G}^{2}$, $\eta =h(D_{ID}||PW_{D}||SK||N_{G}^{2})$, $G_{W}^{5}=R_{SG}^{4}⊕PW_{D}$, and $D_{TID}^{new}=R_{SG}^{4}⊕D_{ID}$. Lastly, the gateway stores $\left\{R_{SG}^{4},D_{TID}^{new}\right\}$ in its secure database and sends a message $\left\{\mu ,SK_{U},\eta ,G_{W}^{5}\right\}$ to the smart device of the doctor.

**Step 5:** After receiving $\left\{\mu ,SK_{U},\eta ,G_{W}^{5}\right\}$ from the gateway, the doctor computes $N_{G}^{2}=\mu ⊕D_{ID}$ and checks the freshness of $N_{G}^{2}$. Then, the smart device computes the session key $SK=\left(SK_{U}⊕N_{G}^{2}\right)⊕PW_{D}$ and $\phi =h(D_{ID}||PW_{D}||SK||N_{G}^{2})$, and verifies $\phi \overset{?}{=}\eta$. If it is equal, the smart device computes $R_{SG}^{4}=G_{W}^{5}⊕PW_{D}$ and $D_{TID}^{new}=R_{SG}^{4}⊕D_{ID}$, and stores $\left\{R_{SG}^{4},D_{TID}^{new}\right\}$ in its memory.

## 5. Cryptanalysis of Masud et al.’s Scheme

If an adversary A obtains a legitimate user’s smart device, A can extract the information $\left\{\beta ,R_{SG}^{1*},D_{TID}\right\}$ from the smart device using a power analysis attack [28], according to Section 3.2. With this information, A can perform various security attacks, such as offline password guessing, user impersonation, and privileged insider attacks. Furthermore, Masud et al.’s scheme does not ensure user anonymity and has a device update problem when signing in for the next session. Details are shown as below.

### 5.1. User Anonymity

An adversary A obtains the smart device of a doctor and extracts $\left\{\beta ,R_{SG}^{1*},D_{TID}\right\}$ using power analysis attack. Then, A calculates $D_{ID}=D_{TID}⊕R_{SG}^{1*}$, where $D_{ID}$ is the real identity of the doctor. Therefore, Masud et al.’s scheme cannot ensure user anonymity.

### 5.2. Offline Password Guessing Attack

An offline password guessing attack has a purpose of obtaining the valid password for a user using a password dictionary in polynomial time. Thus, an adversary A needs some information about the user in order to check whether the guessed password is correct or not. In Masud et al.’s scheme, A can verify the correctness of the guessed password using the information extracted from the smart device of the doctor. We describe the procedures as follows.

**Step 1:** The adversary A inputs a guessed password $PW_{A}$ and calculates $Q^{*}=h(PW_{A}\left|\right|$$R_{SG}^{1*})⊕D_{TID}$.

**Step 2:**A compares $Q^{*}\overset{?}{=}\;\beta$, where $\beta =h(PW_{D}\left|\right|R_{SG}^{1*})⊕D_{TID}$ is a parameter extracted from the smart device of the doctor. If it is equal, it means that A has guessed the password $PW_{D}$ correctly.

Thus, Masud et al.’s scheme is vulnerable to offline password guessing attacks.

### 5.3. User Impersonation Attack

The adversary A can obtain the real identity $D_{ID}$ and the password $PW_{D}$ of the doctor, according to Section 5.1 and Section 5.2. Then, A can impersonate the doctor with this information. We describe the steps as follows.

**Step 1:**A generates a random nonce $N_{A}^{1}$ and computes $N_{A}^{1*}=N_{A}^{1}⊕PW_{D}$ and $\lambda_{A}=h(R_{SG}^{1*}||PW_{D})$. Then, A sends $\left\{N_{A}^{1*},D_{TID},\lambda_{A},S_{TID}\right\}$ to the gateway.

**Step 2:** After receiving $\left\{N_{A}^{1*},D_{TID},\lambda_{A},S_{TID}\right\}$ from the adversary A, the gateway retrieves $N_{A}^{1}=N_{A}^{1*}⊕PW_{D}$ and checks the freshness of $N_{A}^{1}$. If it is found to be fresh, the gateway verifies $D_{TID}$ and $S_{TID}$ from its database. Then, the gateway computes $\lambda^{*}=h(R_{SG}^{1}||PW_{D})$ and compares $\lambda^{*}\overset{?}{=}\;\lambda_{A}$. If the equation is correct, the gateway generates a random nonce $N_{G}^{1}$ and computes $G_{W}^{1}=N_{G}^{1}⊕S_{TID}$, $G_{W}^{2}=h(R_{SN}^{1}\left|\right|R_{SG}^{2})$, $SK_{S}=\left(SK⊕R_{SN}^{1}\right)⊕N_{G}^{1}$ and $G_{W}^{3}=R_{SG}^{3}⊕R_{SN}^{1}$. Finally, the gateway sends $\left\{G_{W}^{1},G_{W}^{2},D_{TID},SK_{S},G_{W}^{3}\right\}$ to the sensor node.

**Step 3:** The sensor node receives $\{G_{W}^{1},G_{W}^{2},D_{TID},$$SK_{S},$$G_{W}^{3}\}$ and retrieves $N_{G}^{1}=G_{W}^{1}⊕S_{TID}$. If $N_{G}^{1}$ is a fresh random nonce, the sensor node computes $S_{N}^{2}=h(R_{SN}^{1}\left|\right|R_{SG}^{2})$ and compares $S_{N}^{2}\overset{?}{=}\;G_{W}^{2}$. The sensor node generates a random nonce $N_{S}^{1}$ and computes $SK=\left(SK_{S}⊕R_{SN}^{1}\right)⊕N_{G}^{1}$, $S_{N}^{2}=N_{S}^{1}⊕S_{TID}$, $S_{N}^{3}=h(R_{SG}^{2*}\left|\right|R_{SN}^{1}\left||SK\right)$, $S_{N}^{4}=R_{SG}^{2}⊕R_{SN}^{2}$, $R_{S}^{3}G=G_{W}^{3}⊕R_{SN}^{1}$ and $S_{TID}^{new}=R^{3}{)}_{SG}⊕S_{ID}$. The sensor node sends $\left\{S_{N}^{2},S_{N}^{3},S_{N}^{4}\right\}$ and stores $\left\{R_{SN}^{2},R_{SG}^{3},S_{TID}^{new}\right\}$.

**Step 4:** The gateway receives the message $\left\{S_{N}^{2},S_{N}^{3},S_{N}^{4}\right\}$ and retrieves $N_{S}^{1}=S_{N}^{2}⊕S_{TID}$. If $N_{S}^{1}$ is a fresh random nonce, the gateway computes $G_{W}^{4}=h(R_{SG}^{2*}\left|\right|R_{SN}^{1}\left||SK\right)$ and checks $G_{W}^{4}\overset{?}{=}\;S_{N}^{3}$. The gateway computes $R_{SN}^{2}=S_{N}^{4}⊕R_{SG}^{2}$ and $S_{TID}^{new}=R_{SG}^{3}⊕S_{ID}$, and stores $\left\{R_{SN}^{2},R_{SG}^{3},S_{TID}^{new}\right\}$. After this, the gateway generates a random nonce $N_{G}^{2}$ and computes $\mu =D_{ID}⊕N_{G}^{2}$, $SK_{U}=\left(SK⊕PW_{D}\right)⊕N_{G}^{2}$, $\eta =h(D_{ID}||PW_{D}||SK||N_{G}^{2})$, $G_{W}^{5}=R_{SG}^{4}⊕PW_{D}$ and $D_{TID}^{new}=R_{SG}^{4}⊕D_{ID}$. Lastly, the gateway stores $\left\{R_{SG}^{4},D_{TID}^{new}\right\}$ and sends $\left\{\mu ,SK_{U},\eta ,G_{W}^{5}\right\}$ to A.

**Step 5:**A computes $N_{G}^{2}=\mu ⊕D_{ID}$ and verifies the freshness of $N_{G}^{2}$. Then, A computes $SK=\left(SK_{U}⊕PW_{D}\right)⊕N_{G}^{2}$ and $\phi =h(D_{ID}||PW_{D}||SK||N_{G}^{2})$, and compares $\phi \overset{?}{=}\;\eta$. Finally, A computes $R_{SG}^{4}=G_{W}^{5}⊕PW_{D}$ and $D_{TID}^{new}=R_{SG}^{4}⊕D_{ID}$, and stores these parameters $\left\{R_{SG}^{4},D_{TID}^{new}\right\}$.

Therefore, Masud et al.’s scheme cannot prevent an impersonation attack.

### 5.4. Privileged Insider Attack

A privileged insider attack can be performed by an insider adversary A that has unquestioned authority within the system. Therefore, the privileged insider A can obtain various information about users, including registration request messages, and may attempt to calculate the session key or impersonate a legal user.

In Masud et al.’s scheme, a privileged insider adversary A can impersonate a legitimate doctor after obtaining a registration request message $\left\{D_{ID},PW_{D},R_{req}\right\}$ and the secret parameter $\left\{\beta ,R_{SG}^{1*},D_{TID}\right\}$ extracted from the smart device of the doctor. A generates a random nonce $N_{A}^{1}$ and computes $N_{A}^{1*}=N_{A}^{1}⊕PW_{D}$ and $\lambda_{A}=h(R_{SG}^{1*}||PW_{D})$. Then, A sends a message $\left\{N_{A}^{1*},D_{TID},\lambda_{A},S_{TID}\right\}$. The gateway and the sensor node authenticate each other and return a message $\left\{\mu ,SK_{U},\eta ,G_{W}^{5}\right\}$ to A. Lastly, A calculates $N_{G}^{2}=\mu ⊕D_{ID}$ and the session key $SK=\left(SK_{U}⊕N_{G}^{2}\right)⊕PW_{D}$. Thus, Masud et al.’s scheme is insecure against privileged insider attacks.

### 5.5. Device Update Problem

The smart device replaces $\left\{R_{SG}^{1*},D_{TID}\right\}$ with $\left\{R_{SG}^{4},D_{TID}^{new}\right\}$ at the end of the authentication and key agreement phase. After this, the doctor may try to authenticate another sensor node that is attached to a patient in other session. However, the doctor cannot perform the login phase. If the doctor inputs a password $PW_{D}$, the smart device computes $Q=h(PW_{D}\left|\right|R_{SG}^{4})⊕D_{TID}^{new}$ and verifies $Q\overset{?}{=}\;\beta$. Since $\beta =h(PW_{D}\left|\right|R_{SG}^{1*})⊕D_{TID}$, the login phase is aborted. Therefore, Masud et al.’s scheme has a device update problem.

## 6. Proposed Scheme

Although Masud et al.’s scheme has efficiency for WMSNs, their scheme has several security vulnerabilities. To address these security weaknesses, we propose a secure three-factor-based mutual authentication and key agreement scheme using PUF. Our scheme consists of initialization, user registration, sensor node registration, mutual authentication and key agreement, and password change phases.

### 6.1. Initialization Phase

Before starting the registration phase, the gateway inserts an identity and a challenge into the sensor node. Figure 3 shows the initialization of our scheme and detailed steps are as follows.

**Step 1:** The gateway selects an identity $S_{ID}$, a challenge $CH_{1}$, and sends $\left\{S_{ID},CH_{1}\right\}$ to the sensor node via a secure channel.

**Step 2:** The sensor node stores $\left\{S_{ID},CH_{1}\right\}$ in the memory.

### 6.2. User Registration Phase

A doctor must register in the network to provide a convenient remote medical service to patients. We show the sensor node registration phase in Figure 4 and detailed steps are as follows.

**Step 1:** A doctor inputs an identity $D_{ID}$, a password $PW_{D}$, and biometric template $BIO_{D}$ to the smart device. Then, the smart device generates a registration request message $R_{req}$ and computes $Gen\left(BID_{D}\right)=<R_{D},P_{D}>$ and $GPW_{D}=h(PW_{D}\left|\right|R_{D})$, where Gen(.) is a fuzzy extractor generation function. The doctor sends $\left\{D_{ID},GPW_{D},R_{req}\right\}$ to the gateway via a secure channel.

**Step 2:** After receiving $\left\{D_{ID},GPW_{D},R_{req}\right\}$ from the doctor, the gateway generates random numbers $R_{SG}^{1}$ and $R_{SG}^{2}$, and computes $\alpha =h(D_{ID}||s)$, $\beta =\alpha ⊕h(GPW_{D}||R_{SG}^{2})$, $\omega =\alpha ⊕h(R_{SG}^{1}||s)$, and $D_{TID}=h(\alpha ||R_{SG}^{2}\left|\right|R_{SG}^{1})$. The gateway stores $\{\omega ,R_{SG}^{1},$$D_{TID}\}$ in the secure database, and sends $\left\{\beta ,D_{TID},R_{SG}^{2}\right\}$ to the doctor via a secure channel.

**Step 3:** The doctor computes $\theta =h(R_{D}||GPW_{D})⊕R_{SG}^{2}$ and $Ver_{D}=h(D_{ID}\left|\right|GPW_{D}$$\left|\right|R_{D})$, and stores $\left\{\beta ,\theta ,Ver_{D},D_{TID},P_{D}\right\}$ in the memory.

### 6.3. Sensor Node Registration Phase

A patient must register in the network using a sensor node in order to receive remote medical services from the doctor. In Figure 5, we show the sensor node registration phase of our scheme and details are as below.

**Step 1:** The sensor node retrieves the challenge stored in the memory and computes $RE_{1}=PUF\left(CH_{1}\right)$, $Gen\left(RE_{1}\right)=<R_{SN}^{1},P_{SN}>$, and $AS_{ID}=h(R_{SN}^{1}\left|\right|S_{ID})$. Then, the sensor node sends $\left\{S_{ID},AS_{ID},CH_{1}\right\}$ to the gateway through a secure channel.

**Step 2:** The gateway generates $R_{SG}^{3}$ and computes $\delta =h(AS_{ID}||s)$, $S_{TID}=h(\delta ||R_{SG}^{3}$$||AS_{ID})$. After this, the gateway stores $\left\{AS_{ID},S_{TID},CH_{1}\right\}$ in its secure database and sends $\left\{\delta ,S_{TID}\right\}$ to the sensor node via a secure channel.

**Step 3:** Finally, the sensor node deletes the challenge $CH_{1}$ and stores $\left\{\delta ,P_{SN},S_{TID}\right\}$ in its memory.

### 6.4. Mutual Authentication and Key Agreement Phase

The doctor sends a login request message to the gateway and establishes a session key among the doctor, the gateway, and the sensor node. After this, the doctor can perform an accurate diagnosis of the patient. We describe the mutual authentication and key agreement phase in Figure 6 and details are as follows.

**Step 1:** The doctor inputs the identity $D_{ID}$, the password $PW_{D}$, and imprints the biometrics $BIO_{i}$. Then, the smart device computes $R_{D}^{*}=Rep\left(BIO_{D},P_{D}\right)$, $GPW_{D}^{*}=h(PW_{D}\left|\right|R_{D}^{*})$, and $Ver_{D}^{*}=h(D_{ID}||GPW_{D}^{*}\left|\right|R_{D}^{*})$, and verifies $Ver_{D}\overset{?}{=}Ver_{D}^{*}$. If it is correct, the smart device generates a random nonce $N_{D}^{1}$ and computes $R_{SG}^{2}=\theta ⊕h(R_{D}||GPW_{D})$, $\alpha =\beta ⊕h(GPW_{D}||R_{SG}^{2})$, $M_{D1}=N_{D}^{1}⊕h(D_{TID}\left||\alpha \right)$, and $V_{D1}=h(N_{D}^{1}\left|\right|D_{TID}\left||\alpha |\right|S_{TID})$. The smart device sends $\left\{D_{TID},S_{TID},M_{D1},V_{D1}\right\}$ to the gateway through a public channel.

**Step 2:** When the gateway receives the message $\left\{D_{TID},S_{TID},M_{D1},V_{D1}\right\}$ from the doctor, the gateway checks the pseudo identity $\left\{D_{TID},S_{TID}\right\}$ and retrieves $\left\{\omega ,R_{SG}^{1}\right\}$ in the database. Then, the gateway computes $\alpha =\omega ⊕h(R_{SG}^{1}||s)$, $N_{D}^{1*}=M_{D1}⊕h(D_{TID}\left||\alpha \right)$, and $V_{D1}^{*}=h(N_{D}^{1*}\left|\right|D_{TID}\left||\alpha |\right|S_{TID})$. If $V_{D1}\overset{?}{=}V_{D1}^{*}$ is correct, the gateway generates a random nonce $N_{G}^{1}$ and retrieves $\left\{AS_{ID},CH_{1}\right\}$. The gateway computes $\delta =h(AS_{ID}||s)$, $M_{G1}=CH_{1}⊕h(\delta ||D_{TID}\left|\right|S_{TID})$, $M_{G2}=(h(N_{D}^{1}\left||\alpha \right)⊕N_{G}^{1})⊕h(\delta ||D_{TID}||AS_{ID})$, and $V_{G1}=h(\delta ||S_{TID}\left|\right|$$AS_{ID}||(h(N_{D}^{1}\left||\alpha \right)⊕N_{G}^{1})||D_{TID})$. After this, the gateway transmits $\left\{D_{TID},S_{TID},M_{G1},M_{G2},V_{G1}\right\}$ to the sensor node via a public channel.

**Step 3:** The sensor node computes $CH_{1}^{*}=M_{G1}⊕h(\delta ||D_{TID}\left|\right|S_{TID})$, $RE_{1}^{*}$$=PUF(CH_{1}^{*})$, $R_{SN}^{1*}=Rep\left(RE_{1}^{{}^{*}},P_{SN}\right)$, $AS_{ID}^{*}=h(R_{SN}^{1*}\left|\right|S_{ID})$, $(h(N_{D}^{1}\left||\alpha \right)$$⊕N_{G}^{1}{)}^{*}=M_{G2}$⊕h(δ$\left|\right|D_{TID}$$||AS_{ID}^{*})$, and $V_{G1}^{*}=h(\delta ||S_{TID}^{*}||AS_{ID}^{*}||(h(N_{D}^{1}\left||\alpha \right)⊕N_{G}^{1}{)}^{*}\left|\right|D_{TID})$. If the equation $V_{G1}^{*}\overset{?}{=}V_{G1}$ is correct, the sensor node generates a random nonce $N_{S}^{1}$ and computes a new pseudo identity $S_{TID}^{new}=h(\delta ||N_{S}^{1}||AS_{ID})$, a session key $SK=h(h(N_{D}^{1}\left||\alpha \right)⊕N_{G}^{1}⊕N_{S}^{1})$, $M_{S1}=N_{S}^{1}⊕h(\delta ||AS_{ID}||h(N_{D}^{1}\left||\alpha \right)⊕N_{G}^{1})$, and $V_{S1}=h(N_{S}^{1}\left|\right|S_{TID}^{new}\left||SK\right)$. Lastly, the sensor node sends $\left\{M_{S1},V_{S1}\right\}$ to the gateway through a public channel and updates $\left\{S_{TID}\right\}$ to $\left\{S_{TID}^{new}\right\}$.

**Step 4:** After receiving $\left\{M_{S1},V_{S1}\right\}$ from the sensor node, the gateway computes $N_{S}^{1*}=M_{S1}⊕h(\delta ||AS_{ID}||h(N_{D}^{1}\left||\alpha \right)⊕N_{G}^{1})$, the session key $SK^{*}=h(h(N_{D}^{1}\left||\alpha \right)⊕N_{G}^{1}⊕N_{S}^{1*})$, the new pseudo identity of the sensor node $S_{TID}^{new*}=h(\delta ||N_{S}^{1*}||AS_{ID})$, and $V_{S1}^{*}=h(N_{S}^{1*}$$\left|\right|S_{TID}^{new*}$$||SK^{*})$. If the equation $V_{S1}^{*}\overset{?}{=}V_{S1}$ is correct, the gateway computes a new pseudo identity of the doctor $D_{TID}^{new}=h(\alpha ||N_{D}^{1}\left|\right|N_{G}^{1}⊕N_{S}^{1})$, $M_{G3}=\left(N_{G}^{1}⊕N_{S}^{1}\right)⊕h(\alpha ||D_{TID})$, and $V_{G2}=h(N_{G}^{1}⊕N_{S}^{1}\left|\right|D_{TID}^{new}\left||SK\right)$. Then, the gateway sends $\left\{M_{G3},V_{G2}\right\}$ to the doctor and updates $\left\{S_{TID},D_{TID}\right\}$ to $\left\{S_{TID}^{new},D_{TID}^{new}\right\}$.

**Step 5:** The doctor computes ${\left(N_{G}^{1}⊕N_{S}^{1}\right)}^{*}=M_{G3}⊕h(\alpha ||D_{TID})$, $D_{TID}^{new*}=h(\alpha ||$$N_{D}^{1}\left|\right|$${\left(N_{G}^{1}⊕N_{S}^{1}\right)}^{*})$, $SK^{*}=h(h(N_{D}^{1}\left||\alpha \right)⊕{\left(N_{G}^{1}⊕N_{S}^{1}\right)}^{*})$, and $V_{G2}^{*}=h(N_{G}^{1*}\left|\right|N_{S}^{1*}\left|\right|D_{TID}^{new*}\left|\right|$$SK^{*})$ and verifies $V_{G2}^{*}\overset{?}{=}V_{G2}$. If it is correct, the doctor replaces $\left\{D_{TID}\right\}$ with $\left\{D_{TID}^{new}\right\}$ in the smart device.

### 6.5. Password Change Phase

In our scheme, we provide a convenient password update process for the doctor. Detailed steps are as follows.

**Step 1:** A doctor inputs $D_{ID}$, $PW_{D}$, and $BIO_{D}$ to the smart device.

**Step 2:** The smart device computes $R_{D}^{*}=Rep\left(BIO_{D},P_{D}\right)$, $GPW_{D}^{*}=h(PW_{D}\left|\right|R_{D}^{*})$, and $Ver_{D}^{*}=h(D_{ID}||GPW_{D}^{*}\left|\right|R_{D}^{*})$ and verifies $Ver_{D}\overset{?}{=}Ver_{D}^{*}$. If the equation is correct, the smart device demands a new password from the doctor.

**Step 3:** The doctor inputs a new password $PW_{D}^{new}$ to the smart device.

**Step 4:** The smart device computes $GPW_{D}^{new}=h(PW_{D}^{new}\left|\right|R_{D})$, $\beta =\alpha ⊕h(GPW_{D}^{new}$$\left|\right|R_{SG}^{2})$, $\theta =h(R_{D}||GPW_{D}^{new})⊕R_{SG}^{2}$, and $Ver_{D}^{new}=h(D_{ID}$$||GP^{new}W_{D}$$\left|\right|R_{D})$ and updates $\left\{\beta ,\theta ,Ver_{D}\right\}$ to $\left\{\beta^{new},\theta^{new},Ver_{D}^{new}\right\}$.

## 7. Security Analysis

To prove the security features of the proposed scheme, we use BAN logic and the RoR model, which can prove the mutual authentication properties and session key security, respectively. Moreover, we show that our scheme has resistance against man-in-the-middle and replay attacks using AVISPA. Furthermore, we claim that the proposed scheme can prevent various security attacks using informal analysis.

### 7.1. BAN Logic

BAN logic is a well-known formal proof to verify the mutual authentication of a protocol. Therefore, many researchers have used BAN logic to prove the mutual authentication of their schemes [30,31,32,33]. In this section, we prove the mutual authentication of the proposed scheme using BAN logic [9]. The basic notations and descriptions of BAN logic are shown in Table 2.

#### 7.1.1. Rules

The logical rules of BAN logic are as follows.

**1.** Message meaning rule (MMR):
$$\frac{P_{1}\;|\equiv P_{1}\overset{K}{↔}P_{2},\;P_{1}⊲{\left\{M_{1}\right\}}_{K}}{P_{1}\left|\equiv P_{2}\right|∼M_{1}}$$**2.** Nonce verification rule (NVR):
$$\frac{P_{1}\left|\equiv \#\left(M_{1}\right),\;P_{1}\right|\equiv P_{2}\;|∼M_{1}}{P_{1}\left|\equiv P_{2}\right|\equiv M_{1}}$$**3.** Jurisdiction rule (JR):
$$\frac{P_{1}|\equiv P_{2}⤇M_{1},\;P_{1}\left|\equiv P_{2}\right|\equiv M_{1}}{P_{1}\;|\equiv M_{1}}$$**4.** Belief rule (BR):
$$\frac{P_{1}\;|\equiv \left(M_{1},M_{2}\right)}{P_{1}\;|\equiv M_{1}}$$**5.** Freshness rule (FR):
$$\frac{P_{1}\;|\equiv \#\left(M_{1}\right)}{P_{1}\;|\equiv \#\left(M_{1},M_{2}\right)}$$

#### 7.1.2. Goals

The BAN logic goals of the proposed scheme are as follows. We define the principals DO, GWN, and SN as the doctor, the gateway, and the sensor node, respectively.

**Goal 1:** 

$DO|\equiv DO\overset{SK}{↔}GWN$

**Goal 2:** 

$DO|\equiv GWN|\equiv DO\overset{SK}{↔}GWN$

**Goal 3:** 

$GWN|\equiv DO\overset{SK}{↔}GWN$

**Goal 4:** 

$GWN|\equiv DO|\equiv DO\overset{SK}{↔}GWN$

**Goal 5:** 

$SN|\equiv SN\overset{SK}{↔}GWN$

**Goal 6:** 

$SN|\equiv GWN|\equiv SN\overset{SK}{↔}GWN$

**Goal 7:** 

$GWN|\equiv SN\overset{SK}{↔}GWN$

**Goal 8:** 

$GWN|\equiv SN|\equiv SN\overset{SK}{↔}GWN$



#### 7.1.3. Idealized Forms

In the proposed scheme, there are four messages exchanged through a public channel. We transform these messages into idealized forms. Our scheme’s idealized forms for the messages are as follows:$Message_{1}$: $DO\to GWN:{\left\{N_{D}^{1}\right\}}_{\alpha }$$Message_{2}$: $GWN\to SN:\{N_{G}^{1},h(N_{D}^{1}{\left||\alpha )\right\}}_{\delta }$$Message_{3}$: $SN\to GWN:{\left\{N_{S}^{1}\right\}}_{\delta }$$Message_{4}$: $GWN\to DO:{\left\{N_{G}^{1},N_{S}^{1}\right\}}_{\alpha }$

#### 7.1.4. Assumptions

The assumptions in the proposed scheme are shown below.

$A_{1}$:

$GWN|\equiv \#(N_{D}^{1})$

$A_{2}$:

$GWN|\equiv \#(N_{S}^{1})$

$A_{3}$:

$SN|\equiv \#(h(N_{D}^{1}\left||\alpha )\right)$

$A_{4}$:

$DO|\equiv \#(N_{G}^{1})$

$A_{5}$:

$DO|\equiv GWN⤇(DO\overset{SK}{↔}GWN)$

$A_{6}$:

$GWN|\equiv DO⤇(DO\overset{SK}{↔}GWN)$

$A_{7}$:

$SN|\equiv GWN⤇(SN\overset{SK}{↔}GWN)$

$A_{8}$:

$GWN|\equiv SN⤇(SN\overset{SK}{↔}GWN)$

$A_{9}$:

$DO|\equiv DO\overset{\alpha }{↔}GWN$

$A_{10}$:

$GWN|\equiv DO\overset{\alpha }{↔}GWN$

$A_{11}$:

$SN|\equiv SN\overset{\delta }{↔}GWN$

$A_{12}$:

$GWN|\equiv SN\overset{\delta }{↔}GWN$



#### 7.1.5. BAN Logic Proof

**Step 1:** We can obtain $S_{1}$ from the message $Message_{1}$.
$$S_{1}:\;GWN⊲{\left\{N_{D}^{1}\right\}}_{\alpha }$$

**Step 2:** We can obtain $S_{2}$ from the message meaning rule using $S_{1}$ and $A_{10}$.
$$S_{2}:\;GWN|\equiv DO|∼(N_{D}^{1})$$

**Step 3:** We can obtain $S_{3}$ from the freshness rule using $S_{2}$ and $A_{1}$.
$$S_{3}:\;GWN|\equiv \#(N_{D}^{1})$$

**Step 4:** We can obtain $S_{4}$ from the nonce verification rule using $S_{2}$ and $S_{3}$.
$$S_{4}:\;GWN|\equiv DO|\equiv (N_{D}^{1})$$

**Step 5:** We can obtain $S_{5}$ from the message $Message_{2}$.
$$S_{5}:\;SN⊲\{N_{G}^{1},h(N_{D}^{1}{\left||\alpha )\right\}}_{\delta }$$

**Step 6:** We can obtain $S_{6}$ from the message meaning rule using $S_{5}$ and $A_{11}$.
$$S_{6}:\;SN|\equiv GWN|∼(N_{G}^{1},h(N_{D}^{1}\left||\alpha )\right)$$

**Step 7:** We can obtain $S_{7}$ from the freshness rule using $S_{6}$ and $A_{3}$.
$$S_{7}:\;SN|\equiv \#(N_{G}^{1},h(N_{D}^{1}\left||\alpha )\right)$$

**Step 8:** We can obtain $S_{8}$ from the nonce verification rule using $S_{6}$ and $S_{7}$.
$$S_{8}:\;SN|\equiv GWN|\equiv (N_{G}^{1},h(N_{D}^{1}\left||\alpha )\right)$$

**Step 9:** We can obtain $S_{9}$ from the message $Message_{3}$.
$$S_{9}:\;GWN⊲{\left\{N_{S}^{1}\right\}}_{\delta }$$

**Step 10:** We can obtain $S_{10}$ from the message meaning rule using $S_{9}$ and $A_{12}$.
$$S_{10}:\;GWN|\equiv SN|∼(N_{S}^{1})$$

**Step 11:** We can obtain $S_{11}$ from the nonce verification rule using $A_{2}$ and $S_{10}$.
$$S_{11}:\;GWN\left|\equiv SN\right|\equiv \left(N_{S}^{1}\right)$$

**Step 12:** We can obtain $S_{12}$ and $S_{13}$ from $S_{8}$ and $S_{11}$. SN and GWN can compute the session key $SK=h(h(N_{D}^{1}\left||\alpha \right)⊕N_{G}^{1}⊕N_{S}^{1})$.
$$S_{12}:\;GWN\left|\equiv SN\right|\equiv \left(SN\overset{SK}{↔}GWN\right)\;\;\;\left(\mathrm{Goal}8\right)$$
$$S_{13}:\;SN\left|\equiv GWN\right|\equiv \left(SN\overset{SK}{↔}GWN\right)\;\;\;\left(\mathrm{Goal}6\right)$$

**Step 13:** We can obtain $S_{14}$ and $S_{15}$ from the jurisdiction rule using $S_{12}$ and $A_{8}$, and $S_{13}$ and $A_{7}$, respectively.
$$S_{14}:\;GWN|\equiv \left(SN\overset{SK}{↔}GWN\right)\;\;\;\left(\mathrm{Goal}7\right)$$
$$S_{15}:\;SN|\equiv \left(SN\overset{SK}{↔}GWN\right)\;\;\;\left(\mathrm{Goal}5\right)$$

**Step 14:** We can obtain $S_{16}$ from the message $Message_{4}$.
$$S_{16}:\;DO⊲{\left\{N_{G}^{1},N_{S}^{1}\right\}}_{\alpha }$$

**Step 15:** We can obtain $S_{17}$ from the message meaning rule using $A_{9}$ and $S_{16}$.
$$S_{17}:\;DO|\equiv GWN|∼(N_{G}^{1},N_{S}^{1})$$

**Step 16:** We can obtain $S_{18}$ from the freshness rule using $S_{17}$ and $A_{4}$.
$$S_{18}:\;DO|\equiv \#(N_{G}^{1},N_{S}^{1})$$

**Step 17:** We can obtain $S_{19}$ from the nonce verification rule using $S_{17}$ and $S_{18}$.
$$S_{19}:\;DO|\equiv GWN|\equiv (N_{G}^{1},N_{S}^{1})$$

**Step 18:** We can obtain $S_{20}$ and $S_{21}$ using $S_{4}$ and $S_{19}$. DO and GWN can compute the session key $SK=h(h(N_{D}^{1}\left||\alpha \right)⊕N_{G}^{1}⊕N_{S}^{1})$.
$$S_{20}:\;DO\left|\equiv GWN\right|\equiv \left(DO\overset{SK}{↔}GWN\right)\;\;\;\left(\mathrm{Goal}2\right)$$
$$S_{21}:\;GWN\left|\equiv DO\right|\equiv \left(DO\overset{SK}{↔}GWN\right)\;\;\;\left(\mathrm{Goal}4\right)$$

**Step 19:** We can obtain $S_{22}$ and $S_{23}$ using the jurisdiction rule using $S_{20}$ and $A_{5}$, $S_{21}$, and $A_{6}$, respectively.
$$S_{22}:\;DO|\equiv \left(DO\overset{SK}{↔}GWN\right)\;\;\;\left(\mathrm{Goal}1\right)$$
$$S_{23}:\;GWN|\equiv \left(DO\overset{SK}{↔}GWN\right)\;\;\;\left(\mathrm{Goal}3\right)$$

### 7.2. RoR Model

In this section, we prove that the session key in the proposed scheme is secure, using the Real-or-Random (RoR) model [10]. To apply our scheme into the RoR model, we discuss the basic concepts of participants, adversaries, and queries. There are three participants in our scheme: $P_{User}^{t_{1}}$, $P_{Gateway}^{t_{2}}$, and $P_{Sensor}^{t_{3}}$, where $t_{k}$ is the participant instance of the user, the gateway, and the sensor node. We assume that an adversary A can control the whole network, which intercepts, deletes, inserts, and eavesdrops messages transmitted through a public channel. Moreover, A attempts to attack the network utilizing Execute, CorruptSD, Reveal, Send, and Test queries in the RoR model. Details of the queries are as follows.

$Execute(P_{User}^{t_{1}},P_{Gateway}^{t_{2}},P_{Sensor}^{t_{3}})$: The query Execute is a passive attack. This query explains that A can eavesdrop messages generated by $P_{User}^{t_{1}}$, $P_{Gateway}^{t_{2}}$, and $P_{Sensor}^{t_{3}}$.$CorruptSD(P_{User}^{t_{1}})$: This query is an active attack. By this query, A can obtain sensitive information extracted from the smart device of $P_{User}^{t_{1}}$.$Reveal(P^{t})$: A can reveal the current session key SK.$Send(P^{t},M)$: Using the query Send, A can send a message M to $P_{User}^{t_{1}}$, $P_{Gateway}^{t_{2}}$, and $P_{Sensor}^{t_{3}}$. Moreover, A can receive the return message. Therefore, this query is an active attack.$Test(P^{t})$: If A performs a Test query, an unbiased coin *C* is flipped prior to starting the game. When the session key SK is fresh, A obtains C=1. A also obtains C=0 when the session key is not fresh. Otherwise, A will receive a null value (⊥). If A cannot distinguish between the session key and the random number, we can ensure that the proposed scheme can provide the security of the session key.

#### Security Proof

**Theorem** **1.**
*In the RoR model, an adversary A tries to calculate the session key of the proposed scheme in polynomial time. Let $Adv_{A}\left(P\right)$ be the possibility that A breaks the security of the session key. We define Hash and PUF as the range space of hash function h(.) and PUF function PUF(.), respectively. In addition, we define $q_{h}$, $q_{p}$, and $q_{s}$ as the number of Hash, PUF, and Send queries, respectively. $l_{D}$ is the number of bits in biometric secret key $BIO_{D}$ of the doctor, $C^{\prime }$ and $s^{\prime }$ are the Zipf’s parameter [34].*

$$Adv_{A}\left(P\right)\leq \frac{q_{h}^{2}}{\left|Hash\right|}+\frac{q_{p}^{2}}{\left|PUF\right|}+2max\left\{C^{\prime }q_{s}^{s^{\prime }},\frac{q_{s}}{2^{l_{D}}}\right\}$$



**Proof.** We follow the security proof as performed in [35,36,37]. In our proof, there are five games $Game_{k}$ where k=0,1,2,3,4. We denote $S_{Game_{k}}$ as the winning probability of the adversary A and $Pr[S_{Game_{k}}]$ as the advantage of the $S_{Game_{k}}$.
$Game_{0}$: $Game_{0}$ is the starting game, where the adversary A picks up the random bit *c*. Therefore, we obtain the following:
(1)$$Adv_{A}\left(P\right)=\left|2Pr\left[S_{Game_{0}}\right]-1\right|$$$Game_{1}$: In this game, A performs an eavesdropping attack, which is the Execute query in the RoR model. When obtaining messages $\left\{D_{TID},S_{TID},M_{D1},V_{D1}\right\}$, $\{D_{TID},$$S_{TID},M_{G1}$, $M_{G2},V_{G1}\}$, $\left\{M_{S1},V_{S1}\right\}$, and $\left\{M_{G3},V_{G2}\right\}$, A carries out Test and Reveal queries to distinguish between the session key SK and a random number. To obtain the session key $SK=h(h(N_{D}^{1}\left||\alpha \right)⊕N_{G}^{1}⊕N_{S}^{1})$, A needs $N_{D}^{1}$, $N_{G}^{1}$, and $N_{S}^{1}$, which are random numbers generated by the user (doctor), the gateway, and the sensor node, respectively. α is the shared secret parameter between the gateway and the user. For these reasons, the adversary A cannot compute the session key SK. This means that A does not enhance the probability compared with the $Game_{0}$.
(2)$$\left[Pr\left[S_{Game_{1}}\right]\right]=\left[Pr\left[S_{Game_{0}}\right]\right]$$$Game_{2}$: In $Game_{2}$, the adversary A performs Send and Hash queries. In the message $\left\{D_{TID},S_{TID},M_{D1},V_{D1}\right\}$, $\left\{D_{TID},S_{TID},M_{G1},M_{G2},V_{G1}\right\}$, $\left\{M_{S1},V_{S1}\right\}$, and $\left\{M_{G3},V_{G2}\right\}$, parameters $D_{TID}$, $S_{TID}$, $V_{D1}$, $V_{G1}$, $V_{S1}$, and $V_{G2}$ are masked by the cryptographic one-way hash function, which provides resistance against hash collision. Moreover, random numbers $N_{D}^{1}$, $N_{G}^{1}$, $N_{S}^{1}$, and the hash functions are contained in $M_{D1}$, $M_{G1}$, $M_{G2}$, $M_{G3}$, and $M_{S1}$. Therefore, there is no collision problem when A performs a Hash query. We apply the birthday paradox [38] and obtain the result as follows:
(3)$$|Pr\left[S_{Game_{2}}\right]-Pr\left[S_{Game_{1}}\right]|\leq \frac{q_{h}^{2}}{\left|Hash\right|}$$$Game_{3}$: $Game_{3}$ is similar to $Game_{2}$. A performs Send and PUF queries. As explained in Section 3.3, the physical function PUF(.) has a secure property. Therefore, we can obtain the following inequation:
(4)$$|Pr\left[S_{Game_{3}}\right]-Pr\left[S_{Game_{2}}\right]|\leq \frac{q_{p}^{2}}{\left|PUF\right|}$$$Game_{4}$: In the final game $Game_{4}$, A performs a CorruptSD query and extracts sensitive data $\left\{\beta ,\theta ,Ver_{D},D_{TID},P_{D}\right\}$ from the smart device of the user. A attempts to calculate parameters α and $R_{SG}^{2}$ from $\beta =\alpha ⊕h(GPW_{D}||R_{SG}^{2})$ and $\theta =R_{SG}^{2}⊕h(R_{D}||GPW_{D})$, respectively. Since parameters $R_{D}$ and $GPW_{D}=h(PW_{D}\left|\right|R_{D})$ are composed of the password and biometrics, A must guess these parameters. Therefore, A cannot enhance the probability because guessing the password and biometrics is a computationally infeasible task. According to Zipf’s law [34], we can make the following inequation:
(5)$$|Pr\left[S_{Game_{4}}\right]-Pr\left[S_{Game_{2}}\right]|\leq max\left\{C^{\prime }q_{s}^{s^{\prime }},\frac{q_{s}}{2^{l_{D}}}\right\}$$When the games are completed, the adversary A obtains the guessed bit *c*. Therefore, it is clear that
(6)$$Pr\left[S_{Game_{4}}\right]=\frac{1}{2}$$By (Equation 2) and (Equation 3), we can obtain the following equation:
(7)$$\frac{1}{2}Adv_{A}\left(P\right)=|Pr\left[S_{Game_{0}}\right]-\frac{1}{2}|=|Pr\left[S_{Game_{1}}\right]-\frac{1}{2}|$$We can obtain the following equation using (Equation 6) and (Equation 7):
(8)$$\frac{1}{2}Adv_{A}\left(P\right)=\left|Pr\left[S_{Game_{1}}\right]-Pr\left[S_{Game_{4}}\right]\right|$$Applying the triangular inequality, we obtain the following result:
$$\begin{array}{c} \frac{1}{2}Adv_{A}\left(P\right)=\left|Pr\left[S_{Game_{1}}\right]-Pr\left[S_{Game_{4}}\right]\right|\\ \leq |Pr\left[S_{Game_{1}}\right]-Pr\left[S_{Game_{3}}\right]|\\ +|Pr\left[S_{Game_{3}}\right]-Pr\left[S_{Game_{4}}\right]|\\ \leq |Pr\left[S_{Game_{1}}\right]-Pr\left[S_{Game_{2}}\right]|\\ +|Pr\left[S_{Game_{2}}\right]-Pr\left[S_{Game_{3}}\right]|\\ +|Pr\left[S_{Game_{3}}\right]-Pr\left[S_{Game_{4}}\right]| \end{array}$$
(9)$$\leq \frac{q_{h}^{2}}{2|Hash|}+\frac{q_{p}^{2}}{2|PUF|}+max\left\{C^{\prime }q_{s}^{s^{\prime }},\frac{q_{s}}{2^{l_{D}}}\right\}$$Finally, we obtain the required result multiplying (Equation 9) by 2:$$Adv_{A}\left(P\right)\leq \frac{q_{h}^{2}}{\left|Hash\right|}+\frac{q_{p}^{2}}{\left|PUF\right|}+2max\left\{C^{\prime }q_{s}^{s^{\prime }},\frac{q_{s}}{2^{l_{D}}}\right\}$$
Thus, we have proven Theorem 1.□

### 7.3. AVISPA Simulation

We simulate the proposed scheme using AVISPA [11,12] to analyze the security features of our scheme. AVISPA is a formal verification tool that can detect security vulnerabilities regarding replay and man-in-the-middle attacks. Therefore, various authentication schemes [39,40,41] have been simulated by using AVISPA.

To simulate our protocol, we need to create a code written in the High-Level Protocol Specification Language (HLPSL). The code written in HLPSL is converted to the Intermediate Format (IF) by the translator. Then, the translator inputs the IF into back-ends. AVISPA has four back-ends, named On-the-Fly Model Checker (OFMC), Constraint Logic-based Attack Searcher (CL-AtSe), SAT-based Model Checker (SATMC), and Three Automata based on Automatic Approximations for Analysis of Security Protocol (TA4SP). In this paper, the OFMC and CL-AtSe back-ends are used because these back-ends provide exclusive-OR operations. Lastly, we obtain the Output Format (OF), which is the security analysis result of the protocol. If we obtain a “SAFE” message in the summary of OF, we can consider that the protocol is secure against replay and man-in-the-middle attacks.

#### 7.3.1. HLPSL Specification

In this section, we explain the HLPSL code of our scheme. There are three basic roles in HLPSL: the doctor DO, the gateway GW, and the sensor node SN. With these roles, we describe the session and the environment roles. The goals, the environment, and the session of our scheme written in HLPSL are shown in Figure 7.

We show the role of the doctor in Figure 8. When state 1 starts, the doctor receives a start message and generates the registration request message $R_{req}$. Then, the doctor computes $GPW_{D}$ with his password $PW_{D}$, the biometrics $BIO_{D}$, and sends $\left\{D_{ID},GPW_{D},R_{req}\right\}$ to the gateway via a secure channel. After this, the doctor receives $\left\{\beta ,D_{TID},R_{SG}^{2}\right\}$ from the gateway and computes $Ver_{D}$ and θ in state 2. The doctor stores $\left\{\beta ,\theta ,Ver_{D},D_{TID},P_{D}\right\}$ in the smart device. With these parameters, the doctor sends a login and authentication request message $\left\{D_{TID},S_{TID},M_{D1},V_{D1}\right\}$ to the gateway via a public channel. witness(DOC,GW, $doc\_gw\_n1d,N_{D}^{\prime 1})$ indicates the freshness of $N_{D}^{1}$. When the doctor receives the message $\left\{M_{G3},V_{G2}\right\}$ in state 3, the doctor performs $request(GW,DOC,doc\_gw\_n1s,N_{S}^{\prime 1})$ and request$\left(GW,DOC,doc\_gw\_n1g,N_{G}^{\prime 1}\right)$, which represent the freshness acceptance of the random nonces $N_{G}^{1}$ and $N_{S}^{1}$.

#### 7.3.2. Simulation Result

We perform simulations using the OFMC and CL-AtSe back-ends and show the simulation result of the proposed scheme in Figure 9. If the summary message is “SAFE”, this indicates that the proposed scheme is secure against replay and man-in-the-middle attacks. As with the simulation result shown in Figure 9, both summaries simulated in the OFMC and CL-AtSe back-ends are “SAFE”. Thus, the proposed scheme can prevent replay and man-in-the-middle attacks.

### 7.4. Informal Analysis

In this section, we show the security features of the proposed scheme, including those that protect against offline password guessing, impersonation, replay, man-in-the-middle, physical, cloning, privileged insider, session-specific random number leakage, and verification table leakage attacks. Moreover, the proposed scheme can ensure user anonymity, perfect forward secrecy, and mutual authentication.

#### 7.4.1. User Anonymity

We assume that an adversary A obtains the stolen smart device of a doctor (user) and extracts $\left\{\beta ,\theta ,Ver_{D},D_{TID},P_{D}\right\}$. However, A cannot compute the real identity of the doctor because the pseudo identity of the doctor $D_{TID}$ is masked by the hash function and updated in every session. Since the parameters $\beta =\alpha ⊕h(GPW_{D}||R_{SG}^{2})$ and $\theta =h(R_{D}||GPW_{D})⊕R_{SG}^{2}$ stored in the smart device are masked in the biometric template of the doctor, the A has difficulty in guessing the real identity of the doctor. Hence, A cannot obtain the real identity of the doctor. Therefore, we demonstrate that the proposed scheme can ensure user anonymity.

#### 7.4.2. Offline Password Guessing Attack

A obtains a doctor’s smart device and obtains $\left\{\beta ,\theta ,Ver_{D},D_{TID},P_{D}\right\}$ from the device using a power analysis attack. Then, A attempts to guess the password of the doctor using the extracted parameters. Unfortunately, A cannot guess the password of the doctor because we use the biometrics in the proposed scheme. Since $GPW_{D}=h(PW_{D}\left|\right|R_{D})$, A must guess not only the password $PW_{D}$ but also the biometrics $BIO_{D}$ of the doctor at the same time. Note that $R_{D}$ is the result of the fuzzy extractor, which is expressed as $R_{D}=Rep\left(BIO_{D},P_{D}\right)$. However, this process is a computationally infeasible task. Thus, the proposed scheme can prevent offline password guessing attacks.

#### 7.4.3. Impersonation Attack

Assume that an adversary A tries to impersonate a legitimate doctor using parameters $\left\{\beta ,\theta ,Ver_{D},D_{TID},P_{D}\right\}$, which are stored in the doctor’s device. Then, A attempts to calculate the login request message $\left\{D_{TID},S_{TID},M_{D1},V_{D1}\right\}$. However, A cannot calculate $M_{D1}=N_{D}^{1}⊕h(D_{TID}\left||\alpha \right)$ and $V_{D1}=h(N_{D}^{1}\left|\right|D_{TID}\left||\alpha |\right|S_{TID})$ because A cannot calculate $\alpha =\beta ⊕h(GPW_{D}||R_{SG}^{2})$. Hence, the proposed scheme is secure against impersonation attacks.

#### 7.4.4. Replay Attack

Assume that an adversary A intercepts authentication request messages $\{D_{TID},$$S_{TID},M_{D1},V_{D1}\}$, $\left\{D_{TID},S_{TID},M_{G1},M_{G2},V_{G1}\right\}$, and sends messages to authenticate the gateway and the sensor node at other sessions. However, each entity checks the freshness of $N_{D}^{1}$, $N_{G}^{1}$, and $N_{S}^{1}$, which are random nonces generated by the doctor, the gateway, and the sensor node, respectively. Therefore, the proposed scheme is secure against replay attacks.

#### 7.4.5. Man-in-the-Middle Attack

We show that A cannot generate the login request message $\left\{D_{TID},S_{TID},M_{D1},V_{D1}\right\}$, according to Section 7.4.3. Moreover, A cannot compute $\left\{D_{TID},S_{TID},M_{G1},M_{G2},V_{G1}\right\}$, $\left\{M_{S1},V_{S1}\right\}$, and $\left\{M_{G3},V_{G2}\right\}$ because each message is masked in the shared secret parameter α and δ. Thus, the proposed scheme can prevent man-in-the-middle attacks.

#### 7.4.6. Physical and Cloning Attacks

We can assume that A physically captures a sensor node $SN_{1}$ and tries to authenticate the gateway as $SN_{1}$. To do this, A obtains the parameters of $SN_{1}$$\left\{\delta ,P_{SN},S_{TID}\right\}$ using a power analysis attack. Then, A attempts to authenticate as a legitimate sensor node $SN_{1}$ using parameters $\left\{\delta ,P_{SN},S_{TID}\right\}$ or by cloning the sensor node $SN_{1}$. When A receives $\left\{D_{TID},S_{TID},M_{G1},M_{G2},V_{G1}\right\}$ from the gateway, A computes $CH_{1}^{*}=M_{G3}⊕h(\delta ||D_{TID}\left|\right|S_{TID})$. However, A cannot compute $RE_{1}$ because the function PUF(.) is a physically unclonable circuit and cannot duplicate, according to Section 3.3. Therefore, A cannot compute $R_{SN}^{1}=Rep\left(RE_{1},P_{SN}\right)$ and $AS_{ID}=h(R_{SN}^{1}\left|\right|S_{ID})$ to calculate $M_{S1}$ and $V_{S1}$. Thus, the proposed scheme is secure against physical and cloning attacks.

#### 7.4.7. Privileged Insider Attack

Assume that a privileged insider A obtains the registration request message $\{D_{ID},$$GPW_{D},R_{req}\}$ of a doctor and obtains parameters $\left\{\beta ,\theta ,Ver_{D},D_{TID},P_{D}\right\}$, extracted from the stolen smart device of the doctor using a power analysis attack, and A attempts to impersonate as the doctor. To compute the login request message $\left\{D_{TID},S_{TID},M_{D1},V_{D1}\right\}$, A must calculate the shared secret parameter α. However, A cannot calculate $\alpha =h(GPW_{D}||R_{SG}^{2})$ because the parameter $R_{D}$ in $GPW_{D}=h(PW_{D}\left|\right|R_{D})$ is generated by the biometrics of the doctor. Moreover, A must guess the password $PW_{D}$ of the doctor to calculate $GPW_{D}=h(PW_{D}\left|\right|R_{D})$, and it is a computationally infeasible task to guess $R_{D}$ and $PW_{D}$ at the same time. Therefore, the proposed scheme can prevent privileged insider attacks.

#### 7.4.8. Session-Specific Random Number Leakage Attack

Suppose that A obtains random nonces $N_{D}^{1}$, $N_{G}^{1}$, and $N_{S}^{1}$. Then, A tries to calculate the session key $SK=h(h(N_{D}^{1}\left||\alpha \right)⊕N_{G}^{1}⊕N_{S}^{1})$. However, A cannot compute the session key SK without knowing the shared secret parameter α. Since α is masked by the hash functions, A cannot calculate α. Thus, the proposed scheme has resistance against session-specific random number leakage attacks.

#### 7.4.9. Verification Table Leakage Attack

If A obtains the verification table $\left\{\omega ,R_{SG}^{1},D_{TID}\right\}$, $\left\{AS_{ID},S_{TID},CH_{1}\right\}$ of the gateway, A attempts to calculate the session key SK or impersonate a doctor. However, A cannot calculate the shared secret parameter $\alpha =\omega ⊕h(R_{SG}^{1}||s)$ and $\delta =h(AS_{ID}||s)$ without the master key *s* of the gateway. Therefore, it is difficult for A to compute the session key $SK=h(h(N_{D}^{1}\left||\alpha \right)⊕N_{G}^{1}⊕N_{S}^{1})$ or impersonate a doctor. Therefore, the proposed scheme can prevent verification table leakage attacks.

#### 7.4.10. Perfect Forward Secrecy

If A obtains the master key *s* of the gateway, A attempts to compute the session key $SK=h(h(N_{D}^{1}\left||\alpha \right)⊕N_{G}^{1}⊕N_{S}^{1})$. However, A cannot compute $\alpha =h(D_{ID}||s)$ without the real identity of the doctor, and all random nonces are masked by hash functions. Therefore, A cannot calculate SK. For this reason, the proposed scheme ensures perfect forward secrecy.

#### 7.4.11. Mutual Authentication

To ensure mutual authentication, each entity checks the validity of $V_{D1}^{*}\;\overset{?}{=}\;V_{D1}$, $V_{G1}^{*}\;\overset{?}{=}\;V_{G1}$, $V_{S1}^{*}\;\overset{?}{=}\;V_{S1}$, and $V_{G2}^{*}\;\overset{?}{=}\;V_{G2}$. Furthermore, all participants check the freshness of random nonces $N_{D}^{1}$, $N_{G}^{1}$, and $N_{S}^{1}$. When the verification processes are successful, we can demonstrate that the participants of the proposed scheme authenticate each other. Therefore, the proposed scheme ensures mutual authentication.

## 8. Performance

In this section, we compare the security features of the proposed scheme with other related schemes [7,18,19,20,25]. Moreover, we show the communication costs, computation costs, and energy consumption of the proposed scheme.

### 8.1. Security Features Comparison

We present the security features of the proposed scheme compared with related schemes [7,18,19,20,25]. In Table 3, we consider various security attacks and functionalities. The security features and the functionalities are as follows: SP1: resistance against smart device theft attack, SP2: resistance against offline password guessing attack, SP3: resistance against impersonation attack, SP4: resistance against replay attack, SP5: resistance against privileged insider attack, SP6: resistance against physical and cloning attacks, SP7: resistance against session-specific random number leakage attack, SP8: resistance against verification table leakage attack, SP9: ensuring user anonymity, SP10: ensuring perfect forward secrecy, SP11: ensuring mutual authentication, SP12: performing RoR model, SP13: performing AVISPA simulation, SP14: performing BAN logic proof. Therefore, our scheme can provide a secure authentication process compared with [7,18,19,20].

### 8.2. Communication Costs Comparison

In this section, we compare the communication costs of the proposed scheme with existing schemes [7,18,19,20,25]. According to [35], we suppose that the SHA-1 hash digest, identity, random number, PUF challenge–response pair, timestamp, and ECC point are 160, 160, 128, 128, 32, and 320 bits, respectively. Therefore, the communication costs of the proposed scheme can be described as follows.

Message 1: The message $\left\{D_{TID},S_{TID},M_{D1},V_{D1}\right\}$ requires (160+160+160+160)=640 bits.Message 2: The message $\left\{D_{TID},S_{TID},M_{G1},M_{G2},V_{G1}\right\}$ needs (160+160+160+160+160)=800 bits.Message 3: The message $\left\{M_{S1},V_{S1}\right\}$ needs (160+160)=320 bits.Message 4: The message $\left\{M_{G3},V_{G2}\right\}$ requires (160+160)=320 bits.

Therefore, the total communication costs of our scheme are 640+800+320+320=2080 bits. In Table 4, we show the total communication costs of our scheme and other related schemes. Consequently, we demonstrate that our scheme has more efficient communication costs than other related schemes [7,18,19,20,25].

### 8.3. Computation Costs Comparison

We compare the computation costs of the proposed scheme with [7,18,19,20,25]. According to [42,43], we define $T_{RNG}$, $T_{H}$, $T_{EM}$, $T_{EA}$, $T_{F}$, and $T_{PUF}$ as the random number generation (≈0.0539 s), hash function (≈0.00023 s), ECC multiplication (≈0.2226 s), ECC addition (≈0.00288 s), fuzzy extractor (≈0.268 s), and PUF operation time (≈0.012 s), respectively. Furthermore, we ignore the execution time of exclusive-OR (⊕) operations because it is computationally negligible.

The total computation costs of our scheme are slightly higher than those of Masud et al.’s scheme [7] as shown in Table 5. However, our scheme has a much higher security level than [7] using the fuzzy extractor and PUF. Moreover, our scheme is more efficient and lightweight than previous schemes [18,19,20,25] that utilize ECC, the fuzzy extractor, and PUF.

### 8.4. Energy Consumption Comparison

In this section, we compare the energy consumption of our scheme with [7,18,19,20,25]. We follow the battery consumption model used in [44], where the energy consumption for sending and receiving a bit are taken as 4.602 mJ and 2.34 mJ, respectively [45]. Therefore, the total energy consumption of our scheme is 4867 mJ. Table 6 shows the total energy consumption of the proposed scheme and [7,18,19,20,25]. The result indicates that our scheme is more efficient in terms of energy consumption than other related schemes.

## 9. Conclusions

In this paper, we review Masud et al.’s scheme and prove that their scheme is vulnerable to offline password guessing, impersonation, and privileged insider attacks. We also discover that Masud et al.’s scheme cannot ensure user anonymity and has a device update problem. To improve the security level and overcome the security weaknesses of Masud et al.’s scheme, we propose a provably secure three-factor-based mutual authentication and key agreement scheme for WMSNs. Our scheme has light weight, using only hash functions and exclusive-OR operators; it provides a secure login process to the doctor using the fuzzy extractor, and it provides resistance against cloning and physical attacks using PUF. We ensure the mutual authentication utilizing BAN logic and prove the session key security of our scheme using the RoR model. We also show that our scheme offers resistance against replay and man-in-the-middle attacks by utilizing the AVISPA simulation tool. We prove that our scheme is secure against various attacks, including offline password, impersonation, sensor node capture, and verification table leakage attacks, through informal analysis. Furthermore, we demonstrate that our scheme can provide user anonymity, perfect forward secrecy, and mutual authentication. Finally, we estimate the computation costs, communication costs, and energy consumption of our scheme and compare it with other related schemes. Our result shows that the proposed scheme can provide doctors and patients with more secure services for WMSNs. In the future, we will develop and implement our scheme, considering performance evaluation and result analysis, confirming its suitability for practical WMSN environments.

## Figures and Tables

**Figure 1 sensors-21-06039-f001:**
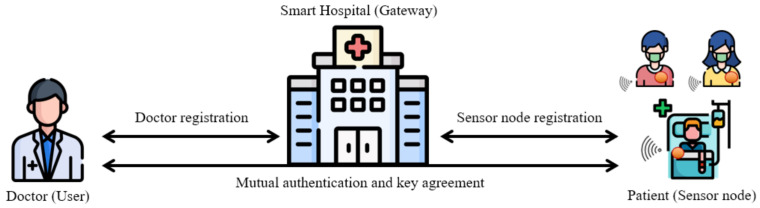
The general system model of WMSNs.

**Figure 2 sensors-21-06039-f002:**
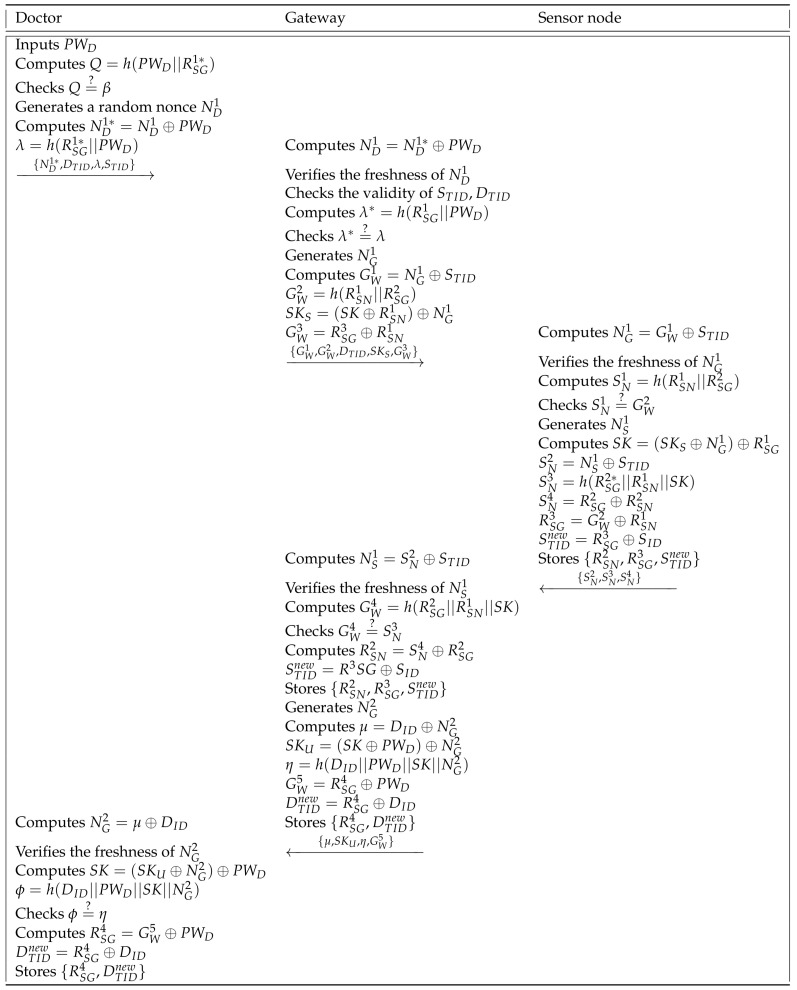
Mutual authentication and key agreement phase of Masud et al.’s scheme.

**Figure 3 sensors-21-06039-f003:**
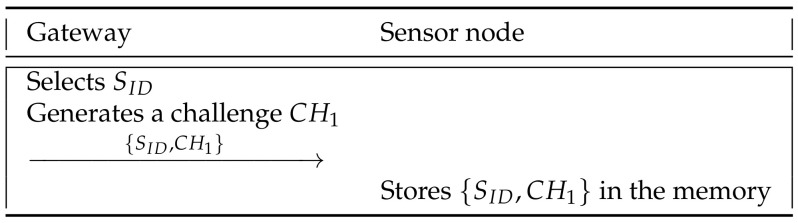
Initialization phase of the proposed scheme.

**Figure 4 sensors-21-06039-f004:**
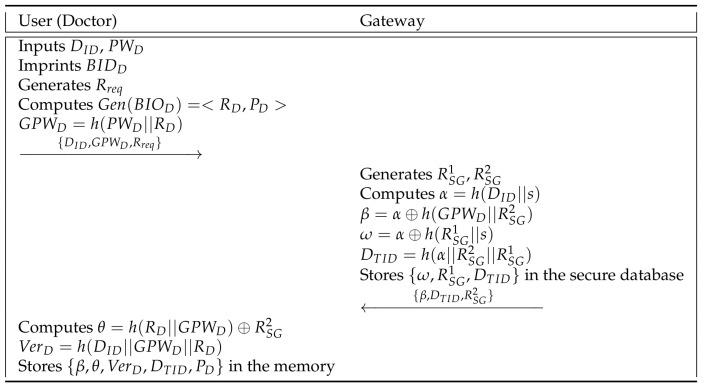
User registration phase of the proposed scheme.

**Figure 5 sensors-21-06039-f005:**
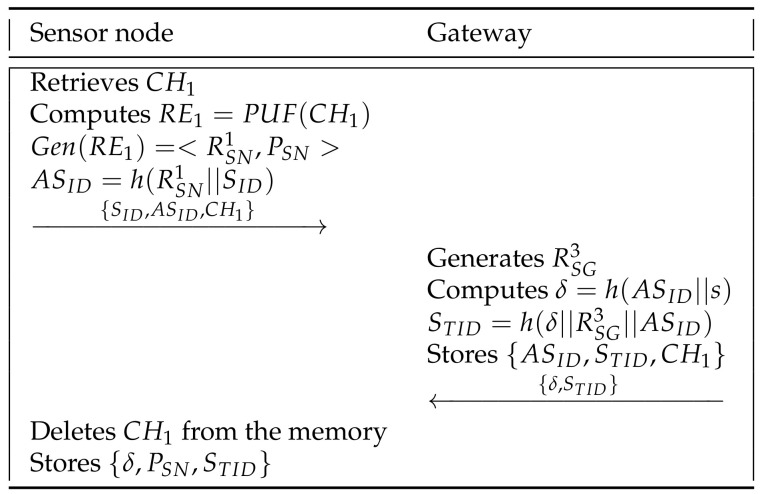
Sensor node registration phase of the proposed scheme.

**Figure 6 sensors-21-06039-f006:**
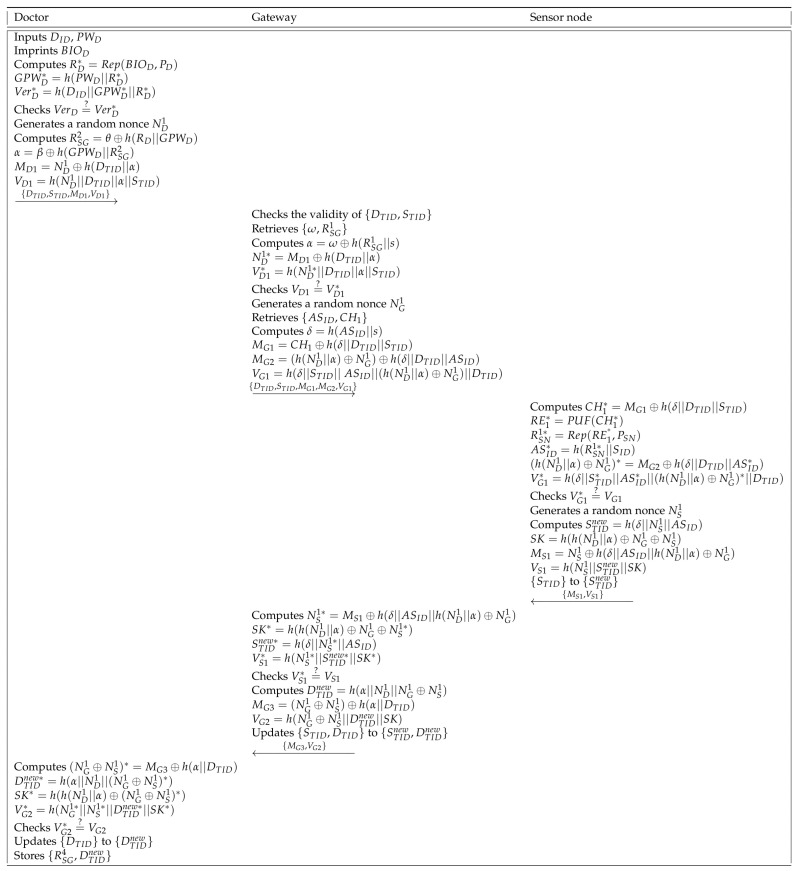
Mutual authentication and key agreement phase of the proposed scheme.

**Figure 7 sensors-21-06039-f007:**
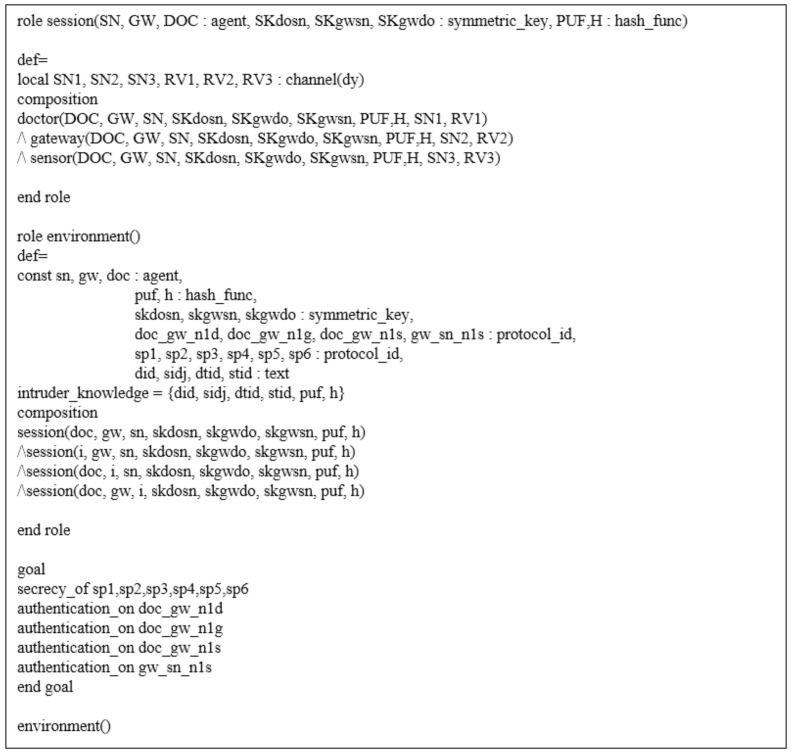
Role specification for the session, environment, and goals.

**Figure 8 sensors-21-06039-f008:**
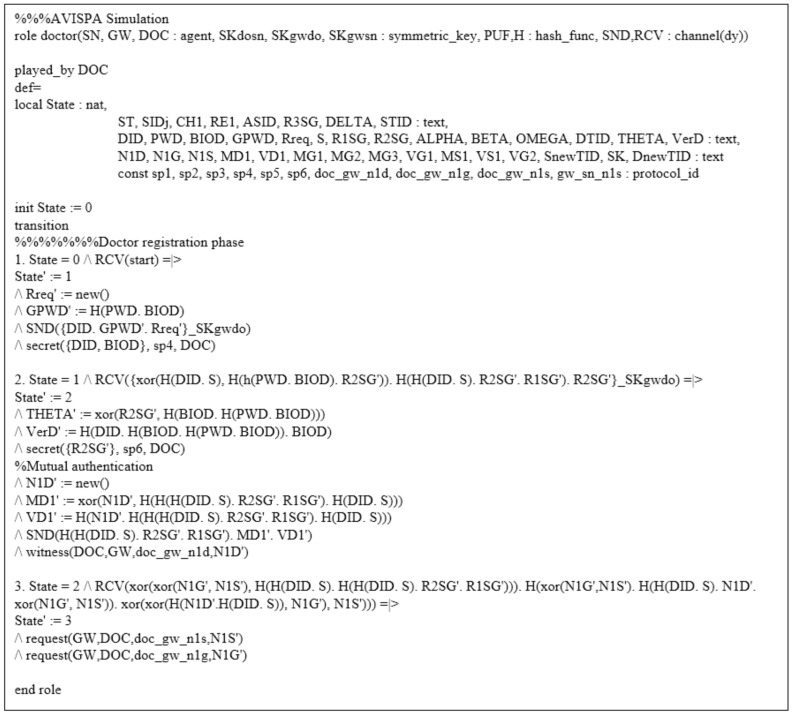
Role specification for the doctor.

**Figure 9 sensors-21-06039-f009:**
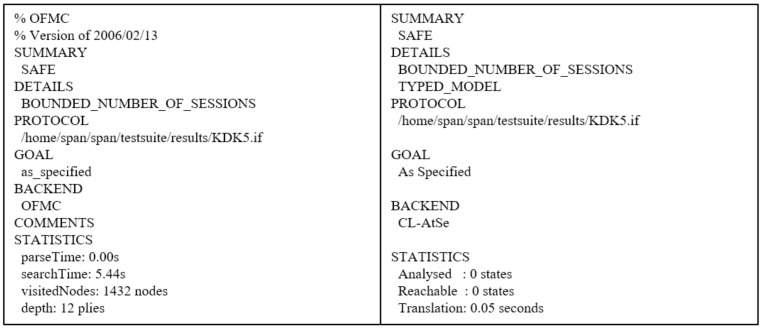
The AVISPA simulation result of the proposed scheme.

**Table 1 sensors-21-06039-t001:** Notations and descriptions.

Notation	Description
$D_{ID},S_{ID}$	Identity of the doctor and the sensor node
$PW_{D}$	Password of the doctor
$BIO_{D}$	Biometric template of the doctor
*s*	Master key of the gateway
$R_{req}$	Registration request message
$R_{SG},R_{SN}$	Random number generated by the gateway and the sensor node
$D_{TID},S_{TID}$	Temporary identity of the doctor and the sensor node
$N_{D},N_{G},N_{S}$	Random nonce generated by device of the doctor, the gateway, and the sensor node
$CH_{1},RE_{1}$	Challenge and response pair
SK	Session key
PUF(.)	Physical unclonable function
h(.)	Hash function
||	Concatenation operator
⊕	Exclusive-OR operator

**Table 2 sensors-21-06039-t002:** Notations of BAN logic.

Notation	Description
$P_{1},P_{2}$	Principals
$M_{1},M_{2}$	Statements
SK	Session key
$P_{1}|\equiv M_{1}$	$P_{1}$ believes $M_{1}$
$P_{1}|∼M_{1}$	$P_{1}$ once said $M_{1}$
$P_{1}⤇M_{1}$	$P_{1}$ controls $M_{1}$
$P_{1}⊲M_{1}$	$P_{1}$ receives $M_{1}$
$\#M_{1}$	$M_{1}$ is fresh
${\left\{M_{1}\right\}}_{K}$	$M_{1}$ is encrypted with *K*
$P_{1}\overset{K}{↔}P_{2}$	$P_{1}$ and $P_{2}$ have shared key *K*

**Table 3 sensors-21-06039-t003:** Security and functionality features comparison.

Security Properties	[18]	[19]	[20]	[25]	[7]	Proposed
SP1	×	🗸	🗸	🗸	×	🗸
SP2	🗸	🗸	🗸	🗸	×	🗸
SP3	🗸	🗸	🗸	🗸	×	🗸
SP4	🗸	🗸	🗸	🗸	🗸	🗸
SP5	×	🗸	🗸	🗸	×	🗸
SP6	×	×	×	🗸	×	🗸
SP7	×	×	🗸	🗸	🗸	🗸
SP8	−	−	−	−	−	🗸
SP9	×	🗸	🗸	🗸	×	🗸
SP10	🗸	🗸	🗸	🗸	🗸	🗸
SP11	🗸	🗸	🗸	🗸	🗸	🗸
SP12	🗸	−	−	🗸	−	🗸
SP13	🗸	🗸	🗸	−	🗸	🗸
SP14	−	🗸	🗸	−	−	🗸

🗸: Provides the security/functionality feature. ×: Does not provide the security/functionality feature. −: Does not consider the security/functionality feature.

**Table 4 sensors-21-06039-t004:** Comparison of communication costs.

Schemes	Total Communication Costs	Messages
Li et al. [18]	2880 bits	4 messages
Shin et al. [19]	3328 bits	4 messages
Ali et al. [20]	2240 bits	4 messages
Chen et al. [25]	2880 bits	5 messages
Masud et al. [7]	2176 bits	4 messages
Proposed	2080 bits	4 messages

**Table 5 sensors-21-06039-t005:** Comparison of computational costs.

Schemes	User	Gateway	Sensor Node	Total	Total Cost (s)
Li et al. [18]	$1T_{RNG}+8T_{H}+3T_{EM}$	$1T_{RNG}+8T_{H}+T_{EM}$	$1T_{RNG}+4T_{H}+2T_{EM}$	$3T_{RNG}+20T_{H}+6T_{EM}$	1.502
Shin et al. [19]	$1T_{RNG}+1T_{F}+14T_{H}+\;2T_{EM}$	$12T_{H}+1T_{EM}$	$1T_{RNG}+5T_{H}+1T_{EM}$	$2T_{RNG}+1T_{F}+31T_{H}+\;4T_{EM}$	1.232
Ali et al. [20]	$1T_{RNG}+1T_{F}+3T_{H}+\;2T_{EM}$	$1T_{RNG}+4T_{H}+2T_{EM}$	$1T_{H}$	$2T_{RNG}+1T_{F}+8T_{H}+\;4T_{EM}$	1.268
Chen et al. [25]	$1T_{RNG}+2T_{F}+14T_{H}+\;1T_{PUF}$	$8T_{H}$	$1T_{RNG}+1T_{F}+8T_{H}$	$2T_{RNG}+3T_{F}+30T_{H}+\;1T_{PUF}$	0.919
Masud et al. [7]	$1T_{RNG}+3T_{H}$	$4T_{RNG}+3T_{H}$	$2T_{RNG}+2T_{H}$	$7T_{RNG}+8T_{H}$	0.379
Proposed	$1T_{RNG}+1T_{F}+11T_{H}$	$1T_{RNG}+15T_{H}$	$1T_{RNG}+1T_{F}+8T_{H}+\;1T_{PUF}$	$3T_{RNG}+2T_{F}+34T_{H}+\;1T_{PUF}$	0.717

**Table 6 sensors-21-06039-t006:** Comparison of energy consumption.

Schemes	Total Energy Consumption
Li et al. [18]	6739 mJ
Shin et al. [19]	7788 mJ
Ali et al. [20]	5242 mJ
Chen et al. [25]	6739 mJ
Masud et al. [7]	5092 mJ
Proposed	4867 mJ

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
