# Peer review of "Provably Secure Three-Factor-Based Mutual Authentication Scheme with PUF for Wireless Medical Sensor Networks"

_sensors, 2021, doi:10.3390/s21186039_

Round 1

Reviewer 1 Report

This article describes wireless medical sensor networks (WMSNs) and presents a method used by the healthcare system to monitor patients. Patients wear a device and the data from that device is sent to doctors via a gateway. Since this data is very important, there must be a secure authentication scheme to protect it.  Masud et al proposed a lightweight and anonymity-preserving user authentication scheme. The authors checked Masud's proposed security scheme and concluded that it cannot provide anonymity to users and they come with a new suitable scheme. More details should be provided on how disruptive technologies like blockchain can be integrated in the methodology

I think that the article it’s very interesting, the main idea is presented clearly, and the details  are presented gradually which makes the information presented easier to understand.
The figure could do with a bit more clarity. These problems are present as well in figure 4, figure 5 and figure 6. Also, the authors should correct the unclarity of the figures showcasing mathematical formulas.

The conclusions are consistent with the research questions and discussion. Future work should be envisioned, especially measurement results from practical implementations.

The main purpose of the article is to present a three-factor-based mutual authentication scheme with physical unclonable functions for wireless medical sensor networks. The authors claim the proposed solution is secure against various security attacks and provides anonymity and secrecy. The scheme is compared to a previous scheme realized by Masud which is claimed to lack resistance to man-in-the-middle attacks, user impersonation and offline password guessing. Various tests presented throughout the paper highlight the proposed solutions superiority to the Masud system. The proposed solution is compared to a previous scheme realized by Masud, but should be compared with other schemes too, especially regarding user impersonation and offline password guessing.  The abstract is very extensive, explaining clearly the context of the research as well as the purpose of the paper. It makes an important point to highlight the comparison done between the proposed system and the Masud et al system. Results of the test are mentioned in the abstract as well as the capabilities of the proposed system.

The article presents a very interesting solution to the problem raised by wireless medical sensor networks. Since these devices require protection of the user data, a protection system is required in order to provide a safe user experience. The proposed system has many benefits and little flaws, as demonstrated by the contents of the paper. A highlight is represented by the ability to secure against security attacks while maintaining anonymity and forward secrecy. The article can provide a very important knowledge base for WMSN developers and doctors who are interested in such devices for their patients.

Future work should be envisioned in the conclusions.

More references to related work regarding other WMSN/IoMT frameworks and blockchain should be added, for example:

- Jan, Saeed Ullah, et al. "Secure Patient Authentication Framework in the Healthcare System Using Wireless Medical Sensor Networks." Journal of Healthcare Engineering 2021 (2021).

- Aileni, Raluca Maria, et al. "IoMT: A blockchain perspective." Decentralised Internet of Things. Springer, Cham, 2020. 199-215.

- Rahman, Mahbubur, and Hamid Jahankhani. "Security Vulnerabilities in Existing Security Mechanisms for IoMT and Potential Solutions for Mitigating Cyber-Attacks." Information Security Technologies for Controlling Pandemics. Springer, Cham, 2021. 307-334.

Reviewer 2 Report

The paper is well written and easy to follow.

The authors present a three-factor mutual authentication scheme using PUF and fuzzy extractor to enhance the security level.

The authors mention the “password guessing” and “privileged insider attacks” from the beginning, but they should define or explain such attacks, for the sake of clarity.

The authors mention in section 3.2. the different types of attacks. This section is clear. However, it is not clear what the goal of the attacker is. For example, what is the gain of an attacker to know the patient’s blood pressure or glucose level? Could the authors please explain and give more details regarding the impact or objective of the attacks?

The paper presents communication and computational costs which are very interesting and accurate. This is a major contribution to the manuscript. However, the communication tasks are related to the data transmission of sensor nodes. In this regard, the energy consumption is not the same for a node that transmits sporadically compared to a node constantly transmitting the information. But the results presented are calculated with no random component. I believe that this is not completely accurate. Specifically, energy and computational cost should be related to the information generation rate and the event duration, since some nodes transmit during a certain event with random duration, while other nodes transmit constantly or periodically. This is particularly important in order to calculate the node lifetime. Or can the authors mention how to calculate the system lifetime with the presented results?
